# Hydantoanabaenopeptins from Lake Kinneret *Microcystis* Bloom, Isolation, and Structure Elucidation of the Possible Intermediates in the Anabaenopeptins Biosynthesis

**DOI:** 10.3390/md21070401

**Published:** 2023-07-13

**Authors:** Shira Weisthal Algor, Assaf Sukenik, Shmuel Carmeli

**Affiliations:** 1Raymond and Beverly Sackler Faculty of Exact Sciences, School of Chemistry, Tel Aviv University, Tel Aviv 69978, Israel; shiraw1@mail.tau.ac.il; 2The Yigal Allon Kinneret Limnological Laboratory, Israel Oceanographic & Limnological Research Institute, Migdal 49500, Israel; assaf@ocean.org.il

**Keywords:** cyanobacteria, *Microcystis*, anabaenopeptin, cyclopeptide, hydantoin derivatives

## Abstract

Anabaenopeptins are common metabolites of cyanobacteria. In the course of reisolation of the known aeruginosins KT608A and KT608B for bioassay studies, we noticed the presence of some unknown anabaenopeptins in the extract of a *Microcystis* cell mass collected during the 2016 spring bloom event in Lake Kinneret, Israel. The ^1^H NMR spectra of some of these compounds presented a significant difference in the appearance of the ureido bridge protons, and their molecular masses did not match any one of the 152 known anabaenopeptins. Analyses of the 1D and 2D NMR, HRMS, and MS/MS spectra of the new compounds revealed their structures as the hydantoin derivatives of anabaenopeptins A, B, F, and ^1^[Dht]-anabaenopeptin A and oscillamide Y (**1**, **2**, **3**, **6**, and **4**, respectively) and a new anabaenopeptin, ^1^[Dht]-anabaenopeptin A (**5**). The known anabaenopeptins A, B, and F and oscillamide Y (**7**, **8**, **9,** and **10**, respectively) were present in the extract as well. We propose that **1**–**4** and **6** are the possible missing intermediates in the previously proposed partial biosynthesis route to the anabaenopeptins. Compounds **1**–**6** were tested for inhibition of the serine proteases trypsin and chymotrypsin and found inactive at a final concentration of ca. 54 μM.

## 1. Introduction

Anabaenopeptins are common metabolites of cyanobacteria, especially water-bloom-forming genera, but have been isolated from terrestrial and marine (*Anabeana* sp. and *Nudularia spumigena*) cyanobacteria and marine sponges (*Theonella* spp. and *Melophlus* sp.) as well [1]. They are nonribosomally synthesized cyclic-peptides that, in cyanobacteria, are composed of five variable L-amino acids and a conserved D-lysine. Their general structure is ^1^X-CO-[^2^Lys-^3^X-^4^X-*N*Me-^5^X-^6^X], where their macrocycle is derived from the cyclization of Lys-e-NH_2_ to the carboxyl of the sixth amino acid. The Lys-a-amine is connected through an ureido bridge to the a-amine of the first amino acid. The chemical shifts of the latter urea-amide protons are distinctive and commonly allocated between 6 and 7 ppm in the ^1^H NMR in DMSO-*d*_6_. The latest report sums the number of known anabaenopeptins to 152, many of which resulted from LC–MS/MS studies, which, according to our findings, sometimes fail to report the correct structure of the compounds (see below) [2]. The anabaenopeptins were reported to exhibit inhibitory activity toward serine proteases, carboxypeptidases, and phosphatases [1]. Their biosynthesis was studied by the analyses of gene clusters isolated from different strains of cyanobacteria, *Anabaena* sp., *Nodularia spumigena* [3], *Nostoc punctiforme* [4], *Sphaerospermopsis torques-reginae* [5], *Planktothrix agardhii* [6], and *Snowella* sp. [7], and three strains of *Microcystis aeruginosa* [8,9], all showing great similarity. However, no explanation was given in these studies to the transformation of the amide bond between the *N*-terminal amino acid and ^2^Lys to an ureido bridge between the a-amines of the latter amino acids. In the process of isolating the known aeruginosins KT608A and KT608B for bioassay studies, from the 2016 spring *Microcystis* bloom (dominant by the *Microcystis aeruginosa* strain marked Mic B due to its dominant pigmentation, brown color) [10,11], we isolated small quantities of 10 anabaenopeptins (**1**–**10**), 6 of which (**1**–**6**) turned to be novel, and their structure elucidation is described herein (Figure 1). Only one of the compounds, **5**, was identified as a new anabaenopeptin derivative. The other five compounds, **1**–**4** and **6**, presented an unusual ^1^H NMR spectrum for anabaenopeptins, where the typical ureido bridge protons were missing from the spectrum, but the distinctive *N*Me signal was present. Four of the latter five compounds presented doubling of some proton and carbon signals in their ^1^H and ^13^C NMR spectra. A full structure elucidation of these compounds revealed that they share a unique hydantoin moiety composed of a urea-type CO, a carboxyamide, a Ca and a-NH of a varied amino acid, and the a-N of the lysine residue. Based on the structure of these compounds, we propose four alternative biosynthetic routes to the anabaenopeptins, which are based on the partial biosynthetic route proposed, established by genome mining, in the past [3,4,5,6,7,8,9].

## 2. Results and Discussion

Hydantoanabaenopeptin A (**1**) presented a molecular ion (negative HR ESI MS, [M-H]^−^) at *m*/*z* 824.4016 corresponding to the molecular formula C_44_H_55_N_7_O_9_ and 21 degrees of unsaturation. The IR spectrum of **1** presented stretching bands at 3274 (amide-NH), 1766 (CO), 1710 (CO), 1640 (amide-CO), and 1592 (aromatic-C=C) cm^−1^, of which those at 1766 (C-4) and 1710 (C-2) cm^−1^ are characteristic of five-membered carbonyls of hydantoin [12]. The ^1^H NMR spectrum of **1** in DMSO-*d*_6_ (see Table 1 for the signals of the major signals and Appendix A for both major and minor signals) presented doubling of signals at a ratio of 2:1 of all of the amide protons and a-protons and many other signals in the spectrum. In the ^13^C NMR spectrum of **1**, most of the carbonyls and α-carbons were doubled at a ratio of 2:1, as well as few of the amino acid side-chain carbons (Appendix A). Analysis of the 2D NMR data (COSY, TOCSY, HSQC, and HMBC) of **1** in DMSO-*d*_6_ revealed the presence of the following amino acid residues: Phe, *N*Me-Ala, Hty, Val, Tyr, and a trisubstituted Lys (in which the expected a-amide proton was missing) and a urea-type carbonyl, similar to the one incorporated in the anabaenopeptin skeleton (Appendix A) [13,14]. The latter composition of amino acids resembled those of anabaenopeptin A (**7**) [13]. The assembly of the amino acids by HMBC and ROESY correlations revealed the same amino acid sequence, cyclo-(Phe-*N*MeAla-Hty-Val-Lys-ε-NH)-, as in anabaenopeptin A, for the macrocycle of **1**. The carbon chemical shifts of Lys C-1 to C-4 were different from the regular chemical shifts of these carbons in the anabaenopeptins [13]. Lys C-1 and C-3 were up-field-shifted (γ substitution effect), while C-2 and C-4 were down-field-shifted (β and δ substitution effects, respectively), in accordance with an extra substitution of the α-nitrogen of Lys. The diastereotopic character of Lys β-protons (Lys-3a and Lys-3b) appears to be much more pronounced in **1** (δ_H_ 2.12 and 0.84, respectively) than in anabaenopeptin A (**7**, δ_H_ 1.28 and 1.18, respectively). The chemical shifts of the carbons of the substituted Tyr residue of **1** presented some differences from those of anabaenopeptin A (**7**) [13]. Although Tyr-CO (Tyr-1) and the ureido-CO presented chemical shifts similar to those of anabaenopeptin A (**7**), Tyr-C-2 to C-4 presented significant differences, with C-2 up-field-shifted and C-3 and C-4 down-field-shifted relative to those of anabaenopeptin A (**7**) [13]. The chemical shifts of Tyr-H-2 (Dd 0.5 ppm) and especially that of Tyr-NH (Dd 2.6 ppm) were significantly down-field-shifted relative to the latter in anabaenopeptin A (**7**). In the HMBC spectrum, Tyr-CO presented correlations with Lys-H-2, Tyr-H-2, Tyr-NH, and Tyr-H-3a, while “urea”-CO presented correlations with Lys-H-2, Tyr-H-2, and Tyr-NH, suggesting an extra hydantoin-type cycle for **1**, in accordance with the 21 degrees of unsaturation. This finding was supported by the strong fragment ion in the MS/MS spectrum of **1**, *m*/*z* 621.3406 ([C_34_H_47_N_5_O_6_]^+^), which resulted from the fragmentation of the C–N bond between Lys-C-2 and Lys-nitrogen (which is part of the hydantoin moiety, Appendix A). The strongest fragment ion in the spectrum, *m*/*z* 649.3352 (B in Figure 1, [C_35_H_47_N_5_O_7_]^+^), for which we suggest a migration process of ureido-CO to Lys-C-2, and the strong M+H^+^-CO fragment ion at *m*/*z* 798.4196 are, as well, in accordance with the proposed hydantoin moiety. We assumed that the doubling in the NMR spectra of **1** resulted from the restricted rotation of the hydantoin moiety around the Lys C–N bond. However, when we tried to confirm it by running the ^1^H NMR spectrum at an elevated temperature, the ratio of the doubled signals changed to 1:1 when the temperature reached 390 °K, and the ratio remained unchanged after lowering the temperature back to room temperature, suggesting that the doubling in the spectra is a result of the epimerization of a chiral center of one of the amino acids in **1**. Application of Marfey’s method [15] coupled with HR ESI LCMS established the absolute configuration of the amino acids as L-NMeAla, L-Hty, L-Phe, L-Val, D-Lys, and 1:1 D,L-Tyr (Appendix A). We suspect that the tyrosine epimerization occurs due to the easy keto/enol tautomerism of the hydantoin moiety during the compound separation [16]. Based on these arguments, structure **1** was proposed for hydantoanabaenopeptin A.

Hydantoanabaenopeptin B (**2**) presented a negative HR ESI MS molecular ion, [M-H]^−^ ion, at *m*/*z* 817.4361 corresponding to the molecular formula C_42_H_58_N_10_O_8_ and 18 degrees of unsaturation. Like for **1**, the ^1^H and ^13^C NMR spectra of **2** presented doubling at a ratio of ca. 1:1 of all of the amide protons, two carbonyl carbons, two a-carbons, and few side-chain carbons (see Appendix A, Table 1 and Appendix A). Analysis of the 2D NMR data (COSY, TOCSY, HSQC, HMBC) of **2** revealed that it is composed of the amino acids Phe, *N*MeAla, Hty, Val, Lys, and Arg, where the last two amino acids presented the most pronounced doubling of proton and carbon signals (Appendix A). The amino acid composition of **2** was thus found to be identical with those of anabaenopeptin B (**8**) [13]. Based on the HMBC and NOE correlations (Appendix A), the structure of **2** was assigned to be the hydantoin derivative of anabaenopeptin B. The proposed structure was supported by the MS/MS fragment ion at *m*/*z* 200.1147 ([C_7_H_14_N_5_O_2_]^+^), which derived from the cleavage of the Lys C–N bond and the preferred allocation of the positive charge on the arginine moiety, and the strong M+H^+^-CO fragment ion at *m*/*z* 791.4575 (Appendix A). As for **1**, the Lys β-protons presented a pronounced diastereotopic difference. The application of Marfey’s method [15], coupled with HR ESI LCMS, established the absolute configuration of the amino acids as L-NMeAla, L-Hty, L-Phe, L-Val, D-Lys, and 1:2 D,L-Tyr (Appendix A), confirming the structure of hydantoanabaenopeptin B as **2**.

Hydantoanabaenopeptin F (**3**) presented a positive HR ESI MS molecular cluster ion, M+H^+^, at *m*/*z* 833.4688 corresponding to the molecular formula C_42_H_60_N_10_O_8_ and 18 degrees of unsaturation. The ^1^H and ^13^C NMR spectra of **3** presented doubling at a ratio of ca. 1:1 of all of the amide protons, three carbonyl carbons, three α-carbons, and a few side-chain carbons (see Appendix A, Table 1 and Appendix A), similar to those of **2**. Analysis of the 2D NMR data (COSY, TOCSY, HSQC, HMBC) of **3** revealed that it is composed of the amino acids Phe, *N*MeAla, Hty, Ile, Lys, and Arg, where the last three amino acids presented the most pronounced doubling of proton and carbon signals (Appendix A). The sequence of the amino acids in the peptide, Phe, *N*MeAla, Hty, Ile, Lys, and Arg, was elucidated based on the correlations from the HMBC and ROESY 2D spectra of **3** (Appendix A) to be identical with that of anabaenopeptin F (**9**) [14] except of the extra correlations of both Lys-H-2 and Arg-2-NH with Arg-CO and ureido-CO, which suggest the closure of a hydantoin moiety similar to that of **2**. The isoleucine moiety was identified by its characteristic ^1^H and ^13^C chemical shifts (5-Me: δ_H_ 0.89 t, δ_C_ 11.8 CH_3_: 6-Me, δ_H_ 0.89 d, δ_C_ 14.8 CH_3_) to be *allo*-isoleucine in accordance with the chemical shifts of L-*allo*-isoleucine in ferintoic acid A [17] (5-Me: δ_H_ 0.88 t, δ_C_ 11.5 CH_3_: 6-Me, δ_H_ 0.87 d, δ_C_ 14.9 CH_3_) and different from those of L-Ile (5-Me: δ_H_ 0.73–0.82 t, δ_C_ 10.3–10.8 CH_3_: 6-Me, δ_H_ 0.88–1.00 d, δ_C_ 14.9–15.5 CH_3_) in anabaenopeptin F [14], lyngbyaureidamide A [18], and schizopeptin 791 [19] and those of D-Ile (5-Me: δ_H_ 0.88 t, δ_C_ 12.3 CH_3_: 6-Me, δ_H_ 0.85 d, δ_C_ 16.5 CH_3_) in mozamide B [20]. The proposed structure was supported by the MS/MS fragment at *m*/*z* 200.1140 ([C_7_H_14_N_5_O_2_]^+^), which derived from the cleavage of the Lys C–N bond and the preferred allocation of the positive charge on the arginine moiety, the complementary (-C_2_H_3_) strong fragment ion at *m*/*z* 611.2669 (Appendix A), and the strong M+H^+^-CO fragment ion at *m*/*z* 805.4730 (Appendix A). The application of Marfey’s method [15] in conjugation with HR ESI LCMS established the absolute configuration of the amino acids as L-NMeAla, L-Hty, L-Phe, L-*allo*Ile, D-Lys, and 1:2 D,L-Arg (Appendix A), verifying the structure of hydantoanabaenopeptin F as **3**.

Hydanto-oscillamide Y (**4**) gave a negative HR ESI MS molecular ion at *m*/*z* 838.4143 corresponding to a molecular formula of C_45_H_57_N_7_O_9_ and 21 degrees of unsaturation. The ^1^H and ^13^C NMR spectra of **4** (Table 1, Appendix A) presented, in contrast to **1**–**3**, a single rotamer; however, the chemical shifts of the protons and carbons of the Lys and the Tyr moieties were shifted similarly to those of **1**. Analysis of the 2D NMR data (COSY, TOCSY, ROESY, HSQC, HMBC, Appendix A) of **4** revealed that it is composed of the amino acids Phe, *N*MeAla, Hty, Ile, Lys, and Tyr as in oscillamide Y (**10**) [21] and established its structure as hydanto-oscillamide Y. The latter structure was supported by a strong fragment ion in the MS/MS spectrum of **4**, *m*/*z* 635.3560 ([C_35_H_49_N_5_O_6_]^+^), which resulted from the fragmentation of the C–N bond between Lys-C-2 and the nitrogen of the hydantoin moiety (Appendix A), and the fragment ion at *m*/*z* 663.3527 ([C_36_H_49_N_5_O_7_]^+^), in accordance with the fragmentations of **1**. The application of Marfey’s method [15] in conjugation with HR ESI LCMS established the absolute configuration of the amino acids as L-NMeAla, L-Hty, L-Phe, L-*allo*Ile, D-Lys, and L-Tyr (Appendix A), confirming the structure of hydanto-oscillamide Y as **4**.

^1^[Dht]-anabaenopeptin A (**5**) presented a positive HR ESI MS protonated cluster ion at *m*/*z* 872.4572 corresponding to a molecular formula of C_46_H_61_N_7_O_10_ and 20 degrees of unsaturation. The latter molecular ion and molecular formula suggest 28 mass units and C_2_H_4_ difference, respectively, from those of anabaenopeptin A. The ^1^H NMR spectrum of **5** in DMSO-*d*_6_ (Table 2, Appendix A) presented a typical anabaenopeptin spectrum [13] with the ureido bridge amide protons appearing as doublets at 6.48 and 6.27 ppm, 13 aromatic protons corresponding to a phenyl and two *p*-phenol rings, a singlet *N*-Me group at 1.76 ppm, and three doublet methyl groups at 1.05, 1.02, and 0.92 ppm, suggesting that **5** contains *N*-MeAla and valine residues. Analysis of the 2D NMR data (COSY, TOCSY, ROESY, HSQC, HMBC, Appendix A) of **5** revealed that the five amino acids of the cycle in it are identical to those of anabaenopeptin A, Phe, *N*MeAla, Hty, Val, and Lys, while the sixth amino acid turned to be an amino acid first described in the anabaenopeptins, which we named dihomotyrosine (Dht) based on the analogy to the known biosynthesis of homophenylalanine and homotyrosine in cyanobacteria [22]. The 2D spectra failed to assign the two carboxy carbons resonating at 174.6 and 171.2 ppm, which were assigned by comparison with those of anabaenopeptin A [13], as Dht-C-1 and Phe-C-1, respectively. The structure of Dht was supported by the fragment ions at *m*/*z* 210.1109 ([C_11_H_16_NO_3_]^+^) and 663.3516 ([C_35_H_45_N_6_O_7_]^+^) in the HR ESI MS/MS of **5** (Appendix A). Dihomotyrosine is homologous to PNV (5-phenylnorvaline) previously described in anabaenopeptins SA4 and SA7 [23]. The sequence of the amino acids in **5** was established as Dht-CO-cyclo-[Lys-Val-Hty-*N*MeAla-Phe] based of the HMBC correlations of Phe-NH with *N*MeAla-CO, *N*MeAla-H-2, and -*N*-Me with Hty-CO, Hty-NH with Val-CO, and the NOE correlations between Dth-NH and Lys-2-NH, Lys-6-NH and Phe-NH, Phe-NH and *N*MeAla-H-2, *N*MeAla-H-2 and Hty-NH, and Hty-H-2 and Val-H-2, leaving only the connection of Val-NH and Lys-CO unproved. However, the strong HR ESI MS/MS fragment ions at *m*/*z* 775.1253, 695.3779, and 637.3714 support the cyclic structure of **5** (Appendix A). The application of Marfey’s method [15] in conjunction with HR ESI LCMS established the absolute configuration of the amino acids as L-NMeAla, L-Hty, L-Phe, L-Val, D-Lys, and L-Dht (Appendix A), establishing the structure of ^1^[Dht]-anabaenopeptin A as **5**.

The anabaenopeptin E [14] structure was established based on analysis of its NMR and MS data to contain 7-methylhomotyrosine at the fourth position. However, MS/MS studies on bloom material or compounds accumulated in animal organs cannot distinguish 7-methylhomotyrosine (MeHty) from dihomotyrosine (due to the identical mass and molecular formula) unless a dipper study based on MS^n^ is utilized. We believe that many of the publications in which MS/MS studies (i.e., Riba et al. [24]) were used to report the structure of cyanobacteria metabolites erroneously report the presence of MeHty instead of Dht.

^1^[Dht]-hydantoanabaenopeptin A (**6**) exhibited a positive HR ESI MS protonated cluster ion at *m*/*z* 854.4464 corresponding to the molecular formula C_46_H_60_N_7_O_9_ and 21 degrees of unsaturation. The ^1^H and ^13^C NMR spectra of **6** (Appendix A) presented a 1:1 signal doubling as in **2**, while the chemical shifts of the protons and carbons of the Lys were shifted similarly to those of **1**. Analysis of the 2D NMR data (COSY, TOCSY, ROESY, HSQC, HMBC, Appendix A) of **6** revealed that it is composed of the same amino acids as **5**—Phe, NMeAla, Hty, Val, Lys, and Dht—and established its structure as ^1^[Dht]-hydantoanabaenopeptin A. The assignment of Dht-C-1 was suggested based on its HMBC correlation with Dht-NH and the NOE correlation of the latter with Dht-H-2. The HMBC correlations of Dht-H-2 and Lys-H-2 with urea-type-CO connected both acids and, in the absence of Lys-a-NH, suggested the hydantoin cycle. The connection of Lys-C-1 to Val-NH was suggested based on the HMBC correlation of Lys-H-2 with Lys-C-1 and the NOE of Val-NH with Lys-H-3a. HMBC correlations of Val-C-1 with Val-H-2 and Hty-NH and of NMeAla-C-1 with NMeAla-H-2 and Phe-NH suggested the connection of the latter amino acids. NMeAla-H-2 and NMe presented correlations with the carboxyl carbon resonating at 171.2 ppm, and NMeAla-H-2 presented NOE correlation with Hty-H-2, assigning the latter carbon as Hty-C-1 and confirming the connection between NMe-Ala and Hty. Lys-e-NH presented HMBC correlation with the carboxyl carbon resonating at 171.1 ppm and NOE correlation with Phe-NH, thus confirming their connection and the assignment of Phe-C-1. The structure of **6** was supported by the characteristic MS/MS fragmentations (as in **1** and **4**): *m*/*z* 826.4515 (M+H^+^-CO, [C_45_H_60_N_7_O_8_]^+^), 649.3719 ([C_35_H_47_N_5_O_7_]^+^), and 622.3248 ([C_34_H_48_N_5_O_6_]^+^) (Appendix A), affirming the hydanto-Dht-[Lys-Val-Hty-NMeAla-Phe] structure of **6**. The absolute configuration of the amino acids of **6** was established by the application of Marfey’s method [15] in conjunction with HR ESI LCMS to be L-NMeAla, L-Hty, L-Phe, L-Val, D-Lys, and D,L-Dht (Appendix A).

Compounds **1** and **4**–**6** were tested at concentrations of 55.1, 54.2, 53.3, and 53.3 μM, respectively, for the inhibition of the serine protease, chymotrypsin, and compounds **2** and **3** were tested at concentrations of 55.6 and 54.6 μM, respectively, for the inhibition of the serine protease and trypsin, but all were found nonactive at those concentrations.

The biosynthetic route proposed by genome mining and gene analyses [3,4,5,6,7,8,9] suggested a synthesis of a linear hexapeptide that contains in the second position a D-lysin, while the other five amino acids are varied L-amino acids of which the amino acid at the fifth position is *N*-methylated. The assembly of the six amino acids is followed by a release of the peptide from the enzyme and cyclization to the mature anabaenopeptin. However, no explanation was given in these studies to the transformation of the amide bond between the *N*-terminal amino acid and the D-^2^Lys to an ureido bridge between the a-amines of the latter two amino acids. Considering our proposal that the hydantoanabaenopeptins **1**–**4** and **6** are intermediates in the biosynthesis of **7**–**10** and **5**, respectively, several alternative routes are possible. The first one, route A in Figure 2, suggests that the carboxylation of the *N*-terminus amine, cyclization to the hydantoin derivative, opening of the hydantoin to the ureido derivative, cleavage of the C-terminus thioester, and cyclization to the mature anabaenopeptin are mediated by the enzymes encoded in the last module of the biosynthetic cluster. Such a route would require the expression of at least one additional condensing enzyme in the last module of the gene cluster. The latter route is less reasonable due to the composition of enzymatic functions encoded in the last module of the biosynthetic cluster, C, A, T, and Te [3,4,5,6,7,8,9]. Routes B–D (Figure 2) suggest loading of a presynthesized *N*-carboxy or *N*-carboxy-phosphate amino acid to the initial module of the biosynthetic cluster, explaining why no additional condensing enzyme is encoded in the biosynthetic cluster. The loading of a presynthesized *N*-carboxy-amino acid to the first adenylation domain of the anabaenopeptin biosynthetic enzyme cluster might explain its low specificity for certain amino acid [3,4,5,6,7,8,9], if the specificity is for the *N*-carboxy-moiety. Such specificity may lead to the production of a set of anabaenopeptins that differ in the starting unit but have a conserved sequence of amino acids in the ring, as in the case of the current study. In route B (Figure 2), except for the loading of *N*-carboxy-amino acid, all of the other steps leading to the ureido bridge are facilitated by the last module of the biosynthetic cluster. In route C (Figure 2), the hydantoin is proposed to be produced by the condensing enzyme encoded in the first module. The hydrolysis of the hydantoin to the ureido product is proposed to happen as the last step of the biosynthesis by the thioesterase (Te). The fourth possible route D suggests that both the cyclization to the hydantoin and its hydrolysis to the ureido moiety occur during the condensation process of lysine to the *N*-carboxy-amino acid. The hydantoin formation and opening might be catalyzed by a single enzyme of the cyclic amidohydrolase family (allantoinase, dihydropyrimidinase, dihydrooratase, hydantoinase, urease, and imidase) [25] that can reversibly catalyze the concession of *N*-carboxy-peptide to the hydantoin and its stereoselective hydrolysis to the ureido-bridge-containing peptide [26]. Alternatively, such conversion might be spontaneous in a similar fashion to the nonenzymatic isomerization of Asp in protein under physiological conditions. The latter isomerization proceeds through a succinimide intermediate and leads to racemization of the Asp and mixture of Asp and iso-linked-Asp, a total of four isomers [27]. However, the possibility that such transformation occurs spontaneously is low due to the stability toward hydrolysis of the hydantoin moiety of **1**–**4** and **6** in the isolation process (acidic aqueous solution) and even at 390 °K in DMSO-*d*_6_. Thus, if such a process occurs in the biosynthesis of the anabaenopeptins, it should occur on the condensation domain (of the *N*-carboxy-L-amino acid, the donor substrate, and L-Lys, the acceptor substrate) possibly with the aid of the donor pantetheinyl arm (PPE) and the catalytic triad of the acceptor condensation domain. The catalysis might prevent the racemization of the donor amino acid prior to the hydantoin opening and explain the L-configuration of the starter amino acid in the anabaenopeptin despite the tendency of the immature hydantoanabaenopeptins, **1**–**4** and **6**, to racemize [28]. A biosynthetic route to the anabaenopeptins that encompass a mechanism similar to the one proposed for the ureido bridge formation in syringolin A is less feasible due to the occurrence, in the anabaenopeptin gene cluster, of two distinct adenylation and condensation domains for the amino acids involved in the construction of the ureido bridge production, unlike the single one for the syringolins [29]. The involvement of the epimerization domain in conversion of the hydantoin moiety to the mature anabaenopeptin is possible as well [30]. However, in the case of a presynthesized *N*-carboxy-amino acid loading to the first adenylation domain and its linking to the thiol through the *N*-carboxy moiety, one cannot exclude a direct biosynthesis of the ureido bridge. In such a case, the *N*-carboxy moiety might be phosphorylated (activated) and transferred to the thiol of the condensation domain and condensed with the amine of Lys to produce the ureido moiety in one biosynthetic step. Iso-linked amino acids are known in cyanobacterial peptides, i.e., the iso-linked Glu, Asp and MeAsp in the microcystins [31]. Further biosynthetic studies, which are beyond our scientific skills, are required in order to conclude on this issue.

## 3. Material and Methods

### 3.1. General Experimental Procedure

Optical rotation values were obtained on a Jasco P-1010 polarimeter at the sodium D line (589 nm). UV spectra were recorded on an Agilent 8453 spectrophotometer (Agilent, Santa Clara, CA, USA). IR spectra were recorded on a Bruker Tensor 27 FT-IR instrument (Bruker, Billerica, MA, USA). NMR spectra were recorded on Bruker Avance III spectrometers (Bruker Karlsruhe, Germany) at 500.13 MHz for ^1^H and 125.76 MHz for ^13^C; chemical shifts were referenced to TMS δ_H_ and δ_C_ = 0 ppm. DEPT, COSY-45, gTOCSY, gROESY, gHSQC, and gHMBC spectra were recorded using standard Bruker pulse sequences. ESI low- and high-resolution mass spectra and MS/MS spectra were recorded on a Waters (Waters, Milford, MA, USA) Xevo G2-XS QTOP instrument equipped with Acquity Hi Class UPLC (binary solvent manager) with an FTN sample manager, column manager, and PDA detector, using a 2.1 × 50 mm BEH C18 (1.7 mm) column and a flow of 0.1–0.3 mL/min. HPLC separations were performed on an Agilent 1100 Series HPLC system (Agilent, Santa Clara, CA, USA).

### 3.2. Biological Material

*Microcystis* biomass, TAU collection number IL-444, was collected in February 2016 from Lake Kinneret, Israel. The cell mass was frozen and lyophilized. A sample of the cyanobacteria is deposited at the culture collection of Tel Aviv University.

### 3.3. Isolation Procedure

The freeze-dried cell mass (515 g) was extracted with 7:3 MeOH:H_2_O (3 × 4 L). The solvent was evaporated to dryness to furnish the crude extract (45 g). Aliquots of the extract were fractionated (10 g in each separation) on an octadecyl-silica flash column (YMC-GEL, ODS, 120A, 4.4 × 6.4 cm), with an increasing concentration of MeOH in H_2_O. The anabaenopeptins were eluted from the column with 8:2 MeOH:H_2_O (fraction 9, 1 g). The latter fraction (A9, 1 g) was separated on a CombiFlash EZ C-18 column (Teledyne ISCO, 15.5 gr HP C18, 250 mg loaded in each separation) using linear gradient conditions from 95% H_2_O to 100% MeCN (at a rate of 1% MeCN/min and a flow of 7 mL/min), resulting in 15 fractions. The fractions were analyzed by NMR and MS and merged into final 10 fractions (B1–B10). Selected fractions from the above were dissolved (20:80 MeCN:H_2_O) and separated on a semipreparative HPLC RP-18 column (YMCPack ODS-AQ, 10 µm, 250 × 20 mm). Fraction B5 (72 mg) was separated under gradient conditions, from 4:1 to 1:1, 0.05% aqueous formic acid/MeCN, at a rate of 0.4% MeCN/min and a flow of 3 mL/min, to yield the known anabaenopeptin B (**8**, 2.1 mg, R_t_ 29.7 min, 4.1 × 10^−4^ % yield from dry cell weight) and the known oscillamide Y (**10**, 3.3 mg, R_t_ 52.4 min, 6.4 × 10^−4^ % yield). Fraction B6 (65 mg) was separated under gradient conditions, from 4:1 to 2:3, 0.05% aqueous formic acid/MeCN, at a rate of 0.3% MeCN/min and a flow of 2.5 mL/min, to yield the known anabaenopeptin F (**9**, 1.5 mg, R_t_ 40.6 min, 2.9 × 10^−4^% yield) as well as the known anabaenopeptin A (**7**, 2.6 mg, R_t_ 63.1 min, 5.0 × 10^−4^% yield). Fraction B7 (55 mg) was separated under gradient conditions, from 3:1 to 2:3, 0.05% aqueous formic acid/MeCN, at a rate of 0.3% MeCN/min and a flow of 2 mL/min, to yield the novel hydantoanabaenopeptin B (**2**, 2.2 mg, R_t_ 22.2 min, 4.3 × 10^−4^% yield), the novel hydantoanabaenopeptin F (**3**, 1.5 mg, R_t_ 30.3 min, 2.9 × 10^−4^% yield), and the novel ^1^[Dht]-anabaenopeptin A (**5**, 1.2 mg, R_t_ 56.8 min, 2.3 × 10^−4^% yield). Fraction B8 (43 mg) was separated under isocratic conditions, 1:3 0.05% aqueous formic acid/MeCN, at a flow of 2 mL/min, to yield the novel hydantoanabaenopeptin A (**1**, 2.1 mg, R_t_ 46.9 min, 4.1 × 10^−4^% yield). Fraction B9 (37 mg) eluted with isocratic system conditions 36:64 0.05% aqueous formic acid/MeCN, at a flow of 2 mL/min, to yield both the novel hydanto-oscillamide Y (**4**, 1.4 mg, R_t_ 54.1 min, 2.7 × 10^−4^% yield) and the novel ^1^[Dht]-hydantoanabaenopeptin A (**6**, 0.9 mg, R_t_ 71.1 min, 1.7 × 10^−4^% yield).

### 3.4. Physical Data of the Compounds

*Hydantoanabaenopeptin A* (**1**): [a]_D_^22^-14.0 (c 0.0019, MeOH); UV (MeOH); λ_max_ (log ε) 202 (4.05), 222 (3.79), 277 (2.93) nm; IR (ATR Diamond) n_max_ 3274, 2929, 2854, 1766, 1710, 1640, 1592, 1438, 1366 cm^−1^; for ^1^H and ^13^C NMR data, see Appendix A; HRESIMS [M–H]^−^, *m*/*z* 824.3985 (calc for C_44_H_54_N_7_O_9_, 824.3983).

*Hydantoanabaenopeptin B* (**2**): [α]_D_^20^-18.7 (c 0.0009, MeOH); UV (MeOH) λ_max_ (log ε) 202 (4.05), 220 (3.64), 277 (2.47) nm; IR (ATR Diamond) n_max_ 3340, 2931, 2855, 1770, 1710, 1640, 1592, 1457,1432, 1367, 1095 cm^−1^; for ^1^H and ^13^C NMR data, see Appendix A; HRESIMS [M–H]^−^, *m*/*z* 817.4367 (calc for C_41_H_57_N_10_O_8_, 817.4361).

*Hydantoanabaenopeptin F* (**3**): [α]_D_^20^-18.6 (c 0.002, MeOH); UV (MeOH) λ_max_ (log ε) 202 (4.05), 222 (3.71), 277 (2.74) nm; IR (ATR Diamond) n_max_ 3327, 2933, 2810, 1772, 1708, 1670, 1609, 1351 cm^−1^; for ^1^H and ^13^C NMR data, see Appendix A; HRESIMS [M–H]^−^, *m*/*z* 833.4690 (calc for C_44_H_63_N_7_O_9_, 833.4687).

*Hydanto-oscillamide Y* (**4**): [α]_D_^20^-20.0 (c 0.0015, MeOH); UV (MeOH) λ_max_ (log ε) 202 (4.05), 224 (3.79), 277 (2.86) nm; IR (ATR Diamond) n_max_ 3295, 2928, 2857, 1768, 1710, 1644, 1595, 1456, 1425, 1367, 1245 cm^−1^; for ^1^H and ^13^C NMR data, see Appendix A; HRESIMS [M–H]^−^, *m*/*z* 838.4143 (calc for C_45_H_56_N_7_O_9_, 838.4140).

*^1^[Dht]-Anabaenopeptin A* (**5**): [α]_D_^22^-15.3 (c 0.0016, MeOH); UV (MeOH) λ_max_ (log ε) 202 (4.05), 223 (4.03), 276 (3.22) nm; IR (ATR Diamond) n_max_ 3318, 2942, 2861, 1658, 1642, 1595, 1515, 1416, 1244 cm^−1^; for ^1^H and ^13^C NMR data, see Appendix A; HRESIMS [M + H]^+^, *m*/*z* 872.4560 (calc for C_46_H_624_N_7_O_10_, 872.4558).

*^1^[Dht]-Hydantoanabaenopeptin A* (**6**): [α]_D_^22^-8.8 (c 0.0011, MeOH); UV (MeOH) λ_max_ (log ε) 202 (4.05), 222 (3.91), 277 (3.03) nm; IR (ATR Diamond) n_max_ 3320, 2930, 2809, 1771, 1710, 1667, 1610, 1352 cm^−1^; for ^1^H and ^13^C NMR data, see Appendix A; HRESIMS [M + H]^+^, *m*/*z* 854.4464 (calc for C_46_H_60_N_7_O_9_, 854.4453).

### 3.5. Determination of the Absolute Configuration of the Amino Acids by Marfey’s Method [15]

Compounds **1**–**6** and authentic samples of anabaenopeptins A (**7**), B (**8**) and F (**9**) and oscillamide Y (**10**) were hydrolyzed in 6 N HCl (1 mL). The reaction mixture was maintained in a sealed glass bomb at 104 °C for 18 h. The acid was removed in vacuo, and the residue was suspended in 250 μL of H_2_O. A solution of 1-fluoro-2,4-dinitrophenyl-5-L-alanine amide (FDAA) in acetone (0.03 M, 20 μL per each amino acid in the peptide) and NaHCO_3_ (1 M, 20 μL per each amino acid) was added to the reaction vessel. The reaction mixture was stirred at 40 °C for 2.5 h in the dark. HCl (2 M, 10 μL per each amino acid) was added to the reaction vessel, and the solution was evaporated in vacuo. The samples of L-FDAA derivatives were analyzed by ESI LC MS. The analysis was performed on a Waters (USA) Xevo G2-XS QTOP instrument equipped with Acquity Hi Class UPLC (binary solvent manager) with an FTN sample manager, column manager, and PDA detector, using a 2.1 × 50 mm BEH C18 (1.7 mm) column and a flow of 0.1–0.3 mL/min. The mobile phase compositions were (A) Water containing 0.1% formic acid and (B) MeCN containing 0.1% formic acid. The elution gradient was as follows: 1 min of 100% A, linear gradient to 30% B over 60 min, and then recycling by linear gradient to 50% B over 10 min and to 100% A over additional 5 min. Samples of 10 μL were injected, and the flow rate was 0.1–0.3 mL/min. The UV detector was set to 340 nm, and the mass spectrometer was operated in both negative and positive ion modes, scanning between 100 and 1000 mass units. The interpretation of the data was conducted after the run on both positive and negative ion modes using Waters MassLynx software (v.4.1).

### 3.6. Protease Inhibition Assays

The samples for biological assays were dissolved in DMSO at a concentration of 1 mg/mL and were tested for inhibition of the proteases at a single concentration: **1** (55.1 mM), **2** (55.6 mM), **3** (54.6 mM), 4 (54.2 mM), **5** (53.3 mM), and **6** (53.3 mM). Assays were performed in a 96-well plate format.

#### 3.6.1. Trypsin

The assay was performed in a Tris buffer (0.6 g Tris HCl in 100 mL H_2_O, pH 7.5). Benzoyl-L-arginine-*p*-nitroanilide hydrochloride (BAPNA), the trypsin substrate, was dissolved at 1 mg/mL in 1:9 DMSO/buffer. The enzyme was dissolved in buffer at 1 mg/mL. To each well were added 100 μL of buffer, 10 μL of enzyme, and 10 μL of sample. The plate was placed in the spectrophotometer at 37 °C. After 5 min, 100 μL of substrate solution was added to each well, and the plate was placed in the spectrophotometer for the kinetic measurement of the absorbance intensity over 30 min at a wavelength of 405 nm.

#### 3.6.2. Chymotrypsin

The assay was performed in a Tris buffer (0.6 gr TRIS HCl/100 mL H_2_O, pH 7.5). The enzyme and the substrate Suc-Gly-Gly-phenylalanine-*p*-nitroanilide (SGGPNA) were dissolved in the buffer at a concentration of 1 mg/mL. To each well were added 100 μL of buffer, 10 μL of enzyme, and 10 μL of sample. Then, the plate was placed in the spectrophotometer for a 5 min incubation at 37 °C. Thereafter, to each well, 100 μL of substrate solution was added, and the plate was placed in the spectrophotometer for the kinetic measurement of the absorbance intensity over 30 min at a wavelength of 405 nm.

## 4. Conclusions

We are monitoring the seasonal blooms of cyanobacteria in Lake Kinneret for almost three decades [32,33,34]. Along the years, the spring *Microcystis* bloom was shifted from green to brown in color (dominant by the *Microcystis aeruginosa* strain marked Mic B due to its dominant pigmentation, brown color) [10,11]. The brown bloom material contains a large amount of the aeruginosins KT608A and KT608B, which we use for an ongoing study on their ecological role. While isolating the aeruginosins for our study, we noticed several unknown masses in the LCMS of some fractions. The latter masses turned to be the new hydantoanabaenopeptins described above. The occurrence of hydantoanabaenopeptins may explain how the hexapeptide, encoded in the biosynthetic genes responsible for the biosynthesis of the anabaenopeptins in cyanobacteria, is converted to the ureido bridge containing anabaenopeptin. We propose several possible routes that include the formation and opening of the hydantoin moiety to the mature anabaenopeptin. The accumulation of the hydantoanabaenopeptins in this bloom material and the fact that it was not noticed in other studies in the past might result from the misfunction of the enzyme responsible for the production and opening of the hydantoin moiety, or misfunction of one of the enzymes that produces the hydantoin, as a by-product. Further genetic and enzymatic studies, which are beyond our capabilities, are needed to clarify this issue.

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
