# Peer review of "Hydantoanabaenopeptins from Lake Kinneret Microcystis Bloom, Isolation, and Structure Elucidation of the Possible Intermediates in the Anabaenopeptins Biosynthesis"

_marinedrugs, 2023, doi:10.3390/md21070401_

Round 1

Reviewer 1 Report

Carmeli et al investigated the anabaenopeptins of the marine cyanobacteria Microcystis., resulted in the isolation of six new compounds together with four known ones. This work was important, which expanded the chemical diversity of anabaenopeptins.

However, I will recommend it after the following revisions were made.

1. It is better to provide the multiplets and coupling constants in Tables 1 and 2.

2. In Figure S11, there was no 1:1 D,L-Tyr observed. Please also check other figures, such as Figure S22.

3. All the know compounds 710 were not briefly introduced in the manuscript.

4. Have these compounds subjected to bioassays?. There was no bioactivities for these compounds reported in the manuscript, which might be not fits the scope of journal ‘Marine Drugs’.

5. The compound numbering for anabaenopeptin F was missing the subsection ‘Isolation Procedure’.

6. Considering the extensive research work of this manuscript, it is better to provide the section Conclusion. And ‘Determination of the Absolute Configuration of the Amino Acids by Marfey’s Method’ could be set as a subsection 3.5.

There are some typo or grammar errors to be corrected, such as the followings:

1. P1L14: ‘The 1H NMR spectrum of some of these compounds’    ‘The 1H NMR spectra of some of these compounds’

2. There were many mistakes in writing the Greek alphabets, which seriously impacted on reading. Some errors were listed as followings: ‘g substitution effect’    γ substitution effect’, ‘b and d substitution effects’    β and δ substitution effects’, ‘dC    δC   ‘dH    δH’, ‘b-protons’    β-protons’, ‘Dd 0.5 ppm’    ‘Δδ 0.5 ppm’

3. P3L94: ‘The Chemical shifts of...’    ‘The chemical shifts of...’

4. P3L110: ‘back to rt’    ‘back to room temperature’

Author Response

Response to reviewers’ comments

I apologize for uploading, in the submission process, the wrong Word version of the manuscript and thus many of the mistakes found by reviewers were already corrected in the PDF file.

Response to reviewer 1:

Many thanks to the reviewer for his comments for improvement of the manuscript.

Comment: 1. It is better to provide the multiplets and coupling constants in Tables 1 and 2.

Reply: When we wrote the manuscript, we tried to incorporate the multiplicity and coupling constants to the tables, however it made the tables almost twice long. Instead, we added to the Supplementary Material for each compound a table with the entire NMR data, which include all of the multiplicities, coupling constants and 2D correlations and directed (in the text) the readers to these tables. We suggest to keep it as is.

Comment: 2. In Figure S11, there was no 1:1 D,L-Tyr observed. Please also check other figures, such as Figure S22.

Reply: The Tyr chromatogram was partially marked – The strongest peak at 9.6 min (L-Tyr-DNP) has a mass of m/z 434.1307 of the Tyr-DNP but it includes a stronger peak of unknown source (contamination in the spectrometer) at a close mass which appear in the chromatogram. The peak at 12.9 min is of D-Tyr-DNP and is smaller in intensity. Both appear in the chromatograms of 1 and 4. It was corrected in the SM. An additional file with the spectra of the reference D,L-Tyr-mono and -di-DNP chromatograms and spectra and similar chromatograms of compound 1 Marfey’s analysis of the with the mono- and di- derivative are included as a PDF file – We do not have explanation for the bigger peak of the L-Tyr-mono-DNP, however the di-DNP derivatives seems similar in area.

Comment: 3. All the know compounds 710 were not briefly introduced in the manuscript.

Reply: Compounds 710 were mentioned in the original text when the structures of their counter partner hydantoin derivatives were discussed. We corrected this issue and they are directly mentioned in the revised text in lines 56-57.

 Comment: 4. Have these compounds subjected to bioassays? There was no bioactivities for these compounds reported in the manuscript, which might be not fits the scope of journal ‘Marine Drugs’.

 Reply: All compounds were tested for inhibition of Serine proteases, but were found not active. Compounds 1, 4, 5 and 6 were tested for inhibition of Chymotrypsin while 2 and 3 for inhibition of Trypsin. Note of it was added to the manuscript.

Comment: 5. The compound numbering for anabaenopeptin F was missing the subsection ‘Isolation Procedure’. And, ‘Determination of the Absolute Configuration of the Amino Acids by Marfey’s Method’ could be set as a subsection 3.5.

Reply: was added.

Comment: 6. Considering the extensive research work of this manuscript, it is

better to provide the section Conclusion.

Reply: The conclusions were added in the end of the manuscript.

The typos and grammar errors were corrected.

Reviewer 2 Report

1. Authors are providing evidence of the occurrences of novel cyclic peptides housing an unprecedented hydratoin moiety. However, the quality of spectra does not allow a deep assessment of the structure description as proposed. The spectra, mainly HMBC spectra, are poor to confirm the structures as proposed. While it can be suspected for some cases, the hydantoin moiety cannot be confirmed from none of the HMBC provided.

2. Always think of highlighting the magnitude of the shift when comparing chemical shifts of two compounds.

3. Does it make sense to provide the chromatogram of the elution of these compounds, assigning the traces to compounds.

4. Since the amide bond could be formed easily from carboxylic acid and amine/amide, it worth clearing up the idea that the hydantoin derivatives 1-4 then 6 could be artefacts of the common anabaenopeptins by a one step extraction then LCMS, avoiding heating

5. The racemisation in compound 1-3 then 6 should be replaced by tautomerism. This phenomenon should be highlighted in Figure 1, as authors draw the compounds, at least where the tautomerism is applicable.

6. The text needs a deep english editing. I highlighted some issues I found. 

7. None of the entities in scheme 1, to be labeled, appears in the text.

8. It is interesting to notice the formation of the new compounds 1-4 then 6 in the MS2 spectra of the known anabaenopeptins, either from the literature or in the data provided by authors. Similar patterns appear with the break down of namalides, analogues of anabaenopeptins. Therefore, the formation of hydantoin derivatives could be non-enzymatic. It becomes hard to believe in the idea that the hydantoin derivatives are intermediates in the biosynthesis of anabaenopeptins although the arguments make sense. However, the contrary is more probable.

Check the MS once again for english conformity. Punctuation for instance are missing in sentences

Author Response

Response to reviewers’ comments

I apologize for uploading, in the submission process, the wrong Word version of the manuscript and thus many of the mistakes found by reviewers were already corrected in the PDF file.

Response to reviewer 2:

Many thanks to the reviewer for his comments for improvement of the manuscript.

Comment: 1. Authors are providing evidence of the occurrences of novel cyclic peptides housing an unprecedented hydratoin moiety. However, the quality of spectra does not allow a deep assessment of the structure description as proposed. The spectra, mainly HMBC spectra, are poor to confirm the structures as proposed. While it can be suspected for some cases, the hydantoin moiety cannot be confirmed from none of the HMBC provided.

Reply: I think that the spectra provided are of good quality, however for such complex compounds one can’t expect to identify all of the correlations from a full spectrum when the total volume of the file is limited. We thus provided all of the correlations in the SM tables. I attached a PDF of relevant HSQC and HMBC expansions of compound 1 to convince you that our structure elucidation is sound.

Comment: 2. Always think of highlighting the magnitude of the shift when comparing chemical shifts of two compounds.

Reply: I am not sure to what you referred, but if you referred to the effect on the lysin proton chemical shifts (lines 99-101, in the revised spectrum), I do not sure about the identity of the protons before and after the changes (in both compounds) and can’t calculate it. Anyway, I do not think that it is relevant to the discussion, there. It was indicated in the text in order to convince that the difference between the anabaenopeptin A and hydantoanabaenopeptin A is in the vicinity of the Lys moiety.

Comment: 3. Does it make sense to provide the chromatogram of the elution of these compounds, assigning the traces to compounds.

Reply: The scheme of the separation of all compounds is summarized in the Experimental Section and include the column properties and retention times of all of the compounds isolated. I do not see what it will add to the manuscript.

Comment: 4. Since the amide bond could be formed easily from carboxylic acid and amine/amide, it worth clearing up the idea that the hydantoin derivatives 1-4 then 6 could be artefacts of the common anabaenopeptins by a one-step extraction then LCMS, avoiding heating.

Reply:  We study cyanobacteria bloom material for about 30 years using the same extraction procedure (7:3 MeOH/water), fractionation on reversed-phase silica (100% water to 100% MeOH or MeCN) and reversed-phase HPLC (eluted with water/MeCN with 0.1% TFA or HCOOH). In the past when we used MeOH instead of MeCN, we have noticed some extant of production methyl esters from the carboxylic acids of the amino acids at position 1 of the anabaenopeptins. We believe that the chromatographic conditions (water and weak acid) do not favor substitution. In the many times we use such conditions in the past we never noticed the presence of hydantoin derivatives, such as those described in the manuscript. I am sorry, but we have used all of this bloom material for the isolation of the compounds thus we cannot check the existence of the compounds in the crude extract.

Comment: 5. The racemisation in compound 1-3 then 6 should be replaced by tautomerism. This phenomenon should be highlighted in Figure 1, as authors draw the compounds, at least where the tautomerism is applicable.

Reply: I think that the result is racemization as we wrote. The process is trough tautomerism of the hydantoin moiety. We changed Figure 1 to reflect the mixture of isomers at the chiral center of the hydantoin moiety.

Comment: 6. The text needs a deep english editing. I highlighted some issues I found.

Reply: As I wrote above, I apologize for uploading, in the submission process, the wrong Word version of the manuscript and thus many of the mistakes found by reviewers were already corrected in the PDF file. However, the manuscript was rechecked and corrected according to the annotated file.

Comment: 7. None of the entities in scheme 1, to be labeled, appears in the text.

Reply: It was mentioned in the text: “The strongest fragment ion in the spectrum, m/z 649.3352 ([C35H47N5O7]+), for which we suggest a migration process of the ureido-CO to Lys-C-2 (Scheme 1) and the strong M+H+-CO fragment ion at m/z 798.4196 are, as well, in accordance with the proposed hydantoin moiety.” – Lines 116-119 in the revised manuscript.

Comment: 8. It is interesting to notice the formation of the new compounds 1-4 then 6 in the MS2 spectra of the known anabaenopeptins, either from the literature or in the data provided by authors. Similar patterns appear with the break down of namalides, analogues of anabaenopeptins. Therefore, the formation of hydantoin derivatives could be non-enzymatic. It becomes hard to believe in the idea that the hydantoin derivatives are intermediates in the

biosynthesis of anabaenopeptins although the arguments make sense. However, the contrary is more probable.

Reply: The loss of water in the MS/MS spectrum of the anabaenopeptins is not surprising since carboxylic acid tend to lose water easily in the mass spectrometer. The product of water loss in the mass spectra of the anabaenopeptins is identical in mass to that of the molecular ion of the hydantoanabaenopeptins but most probably do not have the same structure since we are dealing in this case with “structures” in the excited state after the collision with the energetic ionizing ion! It is possible that the hydantoin formation is non-enzymatic, however, one can’t relay on excited-state-gas-phase reactions to suggest their formation in solution. In order to conclude on the pathway in which the anabaenopeptins are biosynthesized and whether the hydantoins are intermediates in this process or side products remain to be study, as we suggested in our concluding remarks.

Round 2

Reviewer 1 Report

Three errors were observed:

1. P1L23: One word not two words: ‘in active’    ‘inactive’

2. Delete the redundant figure caption on P2L64&65.

3. There was unnecessary line break after ‘Application of Marfey’s method [15]’(P5L117).

However, these could be corrected during the proofreading.

Reviewer 2 Report

Authors did not address my issues. They tried to challenge them with no evidence to support their positions. I still think it worth showing the purification chromatogram with assigned peaks/traces to compounds as they were isolated. It can be introduced in the SI, not necessarily in the MS.

The fact that the new compound masses have been detected in the MS1 of the known anabaenopeptins is the testimony that these compounds are easily made since only in-source losses of easy-to-release molecule are expected in the source. Therefore, the new compounds could be just artifacts.

The NMR are still not supporting the hydantoin function as proposed by authors. The spin systems are not questionable but their arrangements are, mainly that leading to the hydantoin function. Even with new spectra added as affirmed by authors, it is still impossible to confirm this function.

What authors have listed in the table as HMBC interactions should be confirmed by just inspecting the spectra.

Like in Figure 1, entities in scheme 1 should be labelled/numbered and mentioned in the text where applicable, as suggested in my first review.

I continue to think that 'racemisation' is not the right word to call what happen in the new compounds. Epimerisation or  tautomerism work better. Racemisation will require you have gotten both enantiomers, which is not the case here.
